# Evaluating Emotional Eating in Children from the Perspective of Parents: Psychometric Properties of the Parent Version of the Emotional Eating Scale Adapted for Children and Adolescents

**DOI:** 10.3390/nu16173030

**Published:** 2024-09-08

**Authors:** Emma Summers, Christine A. Limbers

**Affiliations:** Department of Psychology and Neuroscience, Baylor University, Waco, TX 76706, USA; emma_summers3@baylor.edu

**Keywords:** emotional eating, Emotional Eating Scale for Children and Adolescents, parent perspectives

## Abstract

Background: Emotional eating is associated with adverse health outcomes in children, including elevated weight status. Currently, there is not a well-validated parent-report measure of emotional eating for young children. This study assessed the reliability and validity of the 10-item parent version of the Emotional Eating Scale Adapted for Children and Adolescents (EES-C) Short-Form. Methods: The participants were 207 parents and 144 children from the southern United States. They completed the parent- and child-report EES-C Short-Form and responded to measures related to child eating behaviors, mood, and gratitude. Results: The parent-report EES-C Short-Form demonstrated good internal consistency reliability (Cronbach’s alpha = 0.94). Test–retest reliability was also supported, as evidenced by a medium correlation (ICC = 0.56, *p* < 0.001) between parent-rated emotional eating across two time points. Additionally, the measure demonstrated a significant correlation with a scale of emotional overeating (*r* = 0.25, *p* < 0.001)—a theoretically related construct. Supporting discriminant validity, the measure was not significantly related to a measure of parent-reported gratitude (*r* = 0.07, *p* = 0.30). A unidimensional model provided good fit for the data (CFI = 0.997, SRMR = 0.046). Conclusions: The results from the current study provide preliminary evidence supporting the reliability and validity of the parent version of the EES-C Short-Form. For the purpose of screening children in school or primary care settings, the EES-C Short-Form may be practical and helpful in identifying children who may be at risk of developing adverse health outcomes or more-severe eating disorder pathology.

## 1. Introduction

Emotional eating is defined as the act of consuming food in response to negative emotions [1]. An individual’s biogenetics, social environment, and psychological functioning are integral in conceptualizing the development and maintenance of emotional eating [2]. Individuals who emotionally eat report a time-limited escape from uncomfortable, painful, or distressing emotions [3]. Consequently, emotional eating is negatively reinforced and maintained by the reduction in emotional discomfort or distress that eating produces [4]. However, eating in the absence of hunger, or overeating, often produces negative emotions [3], which can then re-produce the cycle of emotional eating [5]. Emotion dysregulation plays an important role in the manifestation of emotional eating [6] and has been found to mediate the association between childhood trauma and emotional eating in young and middle-aged adults [7] as well as the association between psychopathological traits and binge eating in bariatric surgery candidates [8]. Emotional eating can result in weight gain and reduced weight loss over time [9,10,11] and is associated with disordered eating behaviors and attitudes [12,13,14].

Self-report measures are the most widely used method for assessing emotional eating [15] and only recently have begun to be adapted from adult measures for use with children and adolescents [16]. Notably, this form of measurement adaption assumes that children and adolescents are able to introspectively report on their motives for eating and their eating behaviors [17]. Self-reporting on an abstract emotional experience requires the ability to both identify and report on complex and unique experiences [18], which is oftentimes difficult for children who have not reached the necessary developmental milestones [19,20]. To determine whether subjective measures of emotional eating function the way they are intended in child populations, it is important to establish the psychometric properties of these instruments.

Limbers et al. developed a 10-item short-form version of the Emotional Eating Scale for Children and Adolescents [21]. The EES-C Short-Form has demonstrated good internal consistency reliability in two studies and a unidimensional factor structure [21,22]. Adolescents who endorsed loss of control eating on a subjective measure endorsed greater levels of emotional eating than adolescents who did not endorse loss of control eating and the EES-C Short-Form was significantly correlated with the subjective measure of loss of control eating (r = −0.26) [22]. In terms of discriminant validity, the EES-C Short-Form was not significantly correlated with a measure of gratitude, suggesting that the construct measured by the EES-C Short-Form is distinct from a theoretically nonrelated construct [22].

While the EES-C Short-Form self-report version has demonstrated strong psychometric properties [21,22], a major limitation of this measure has been the absence of a parent version that could be used with younger children who may not be able to provide a reliable and valid self-report of their emotional eating behaviors [23]. A parent version of the EES-C Short-Form was recently developed with children 5 to 12 years to address this gap in the empirical literature [24]. A preliminary validation of the parent version of the EES-C Short-Form found that the measure demonstrated strong internal consistency reliability [24]. Supporting convergent validity, maternal reports of their child’s emotional eating were significantly correlated with theoretically related constructs: maternal concern about their child’s weight, restricted eating, and monitoring of their child’s food intake on the Child Feeding Questionnaire [24]. A known-groups analysis found that mothers reported significantly higher levels of emotional eating in children who were overweight/obese than normal-weight children [24]. Lastly, consistent with previous findings of the EES-C child self-report version [21], the parent version of the EES-C Short-Form demonstrated a unidimensional model fit, supporting factor validity [24].

Despite data that indicate strong initial psychometric properties of the parent version of the EES-C Short-Form, further validation work of the measure is needed [24]. The EES-C Short-Form parent-report version’s test–retest reliability and discriminant validity have yet to be examined. Additionally, agreement between raters on the EES-C Short-Form, comparing a child’s self-report of their emotional eating to the report of a parent, has not yet been evaluated. The preliminary study that examined the measurement properties of the parent version of the EES-C Short-Form restricted its sample to mothers, limiting the potential utility of the measure, and also relied on maternal reports of their child’s height and weight to calculate body mass index (BMI), which can be inaccurate [24]. Consequently, the purpose of the current study was to evaluate the psychometric properties of a parent-report version of the EES-C Short-Form to measure emotional eating in children between the ages of 5 and 12 years. Specifically, the proposed study aimed to examine the (a) internal consistency, (b) correlation of parent ratings at two time points to assess test–retest reliability, (c) correlation of the total score with a measure of a theoretically related construct as an indicator of convergent validity, (d) correlation of the total score with a measure of a theoretically nonrelated construct as an indicator of discriminant validity, (e) factor structure and its associated model fit as an indicator of structural validity, (f) rate of agreement between child self-reporting and parent ratings of emotional eating, and (g) differences between groups (i.e., children who were obese versus children with healthy weight) on levels of parent- rated emotional eating. The following hypotheses were tested:The parent version of the EES-C Short-Form total score will demonstrate adequate internal consistency reliability, as evidenced by a Cronbach alpha statistic of 0.80 or greater [24,25].The parent version of the EES-C Short-Form total score will demonstrate moderate test–retest reliability, as evidenced by an intraclass correlation coefficient (ICC) of 0.5 or greater [16,26].To support convergent validity, parent version of the EES-C Short-Form total score will be significantly correlated with a theoretically related construct of maladaptive eating behaviors [27,28,29,30].To support discriminant validity, the parent version of the EES-C Short-Form will demonstrate a small, non-significant correlation with a scale measuring gratitude—a theoretically nonrelated construct [22,27].There will be moderate agreement between raters (i.e., child self-report and parent-report) on the EES-C Short-Form total score, indicated by an intraclass correlation coefficient (ICC) between 0.5 and 0.75 [26].The parent version of the EES-C Short-Form total score will produce a unidimensional model with good fit, as indicated by goodness-of-fit indices equal to or exceeding cutoff values for adequate model fit (CFI ≥ 0.95, SRMR ≤ 0.09), supporting structural validity [24,31].Parents of children who are overweight and obese will report significantly higher levels of emotional eating on the EES-C Short-Form total score than parents of healthy-weight children, evidenced by a medium effect size (i.e., Cohen’s d), approximately 0.5 in magnitude [24,32].

## 2. Materials and Methods

The sample consisted of 207 parents and 144 children aged 5 to 12 years. A total of 63 parents completed the EES-C Short-Form via Qualtrics, two weeks following their initial participation, to assess test–retest reliability. Participants were recruited from six elementary and middle schools in the southern United States and did not receive compensation for their participation. Students at participating schools were sent home with a packet including information about the study, a parent consent form, and the parent permission slip. Also included in the packet were the parent-report measures (i.e., demographic questionnaire, parent-report EES-C Short-Form, CEBQ, and Gratitude Moments Questionnaire) to be completed and returned to the school by the child. On the parent consent form, parents were asked to provide an email address or phone number for future contact if they consented to complete a measure virtually two weeks after initial data collection. Participants received an email or text message with the link to complete the parent-report version of the EES-C Short-Form virtually, for the purpose of assessing test–retest reliability, two weeks following the initial administration.

Children were eligible to participate in the study after returning the parent permission form signed by a parent or guardian. Children who returned parent permission forms who wished to participate were brought to an available classroom to complete the child assent form. Each child was asked to participate in height and weight data collection. Children ages 8 to 12 years old were also asked to complete questionnaires (i.e., demographic questionnaire, and EES-C Short-Form). Height and weight data for children were collected in an area that was not populated by others (e.g., outside of the classroom in use). Unique numerical identifiers were used to link parent and children’s responses. All study procedures were approved the Institutional Review Board at Baylor University.

### 2.1. Measures

#### 2.1.1. Emotional Eating Scale for Children and Adolescents Short-Form, Parent and Self-Report

The EES-C Short-Form is an adapted version of the original 25-item Emotional Eating Scale Adapted for Children and Adolescents [16]. The EES-C Short-Form is a 10-item measure, whose items were drawn from the original 25-item adapted scale, and has been validated in samples ages 10 to 19 years [21,22]. The self-report form indicates, “We all respond to different emotions in different ways. Some types of feelings lead people to experience an urge to eat. Please indicate the extent to which the following feelings lead you to feel an urge to eat by checking the appropriate box”. Participants then endorsed the degree to which they felt the urge to eat in respond to emotions (i.e., sadness, jealousy, loneliness, nervousness, anger, guilt, helplessness, disobedience, and not doing enough) using a five-point Likert scale ranging from 1 (no desire to eat) to 5 (very strong desire to eat). The parent-report version utilized the same items, though the language was modified to, “When your child feels sad/disobedient/jealous/lonely/confused/nervous/angry/guilty/helpless/they’re not doing enough, they have no desire to eat/a small desire to eat/a moderate desire to eat/a strong desire to eat/a very strong desire to eat”. In the current study, the self-report scale demonstrated good internal consistency reliability (Cronbach’s alpha = 0.88).

#### 2.1.2. The Children’s Eating Behavior Questionnaire

The Children’s Eating Behavior Questionnaire (CEBQ) is a 35-item measure consisting of 8 subscales (emotional overeating, enjoyment of food, satiety responsiveness, slowness in eating, food responsiveness, desire to drink, food fussiness, and emotional undereating) that is commonly used to assess child eating patterns in pediatric settings [33,34]. The measure asked parents to “Please read the following statements and tick the boxes most appropriate to your child’s eating behavior”. Examples of items are “My child loves food”, “My child eats more when anxious”, and “My child eats more when s/he has nothing else to do”. Response options included Never (1), Rarely (2), Sometimes (3), Often (4), or Always (5). The CEBQ has been validated for children five to twelve years old and has demonstrated adequate internal consistency (Cronbach’s alphas = 0.75 to 0.90) [34]. The CEBQ’s subscales are theoretically related to emotional eating [35]. In the current study, the subscales comprising the CEBQ ranged from poor to good internal consistency reliability (food responsiveness = 0.34, emotional overeating = 0.62, enjoyment of food = 0.89, desire to drink = 0.71, satiety responsiveness = 0.79, slowness in eating = 0.70, emotional undereating = 0.76, food fussiness = 0.53).

#### 2.1.3. Gratitude Moments Questionnaire

The Gratitude Moments Questionnaire is a 10-item parent-report measure of child gratitude, developed by Hussong and colleagues [36]. The questionnaire asked parents to “Think back about the past week. Please indicate how often you saw the following in your child in the past week”. The items ask parents to report on the number of times a child demonstrated gratitude—for example, “My child used good manners without being prompted (i.e., please or thank you)”. Response options included Not at all (0), Once (1), 2–4 times (2), 5–10 times (3), and 11 or more times (4). The measure has demonstrated good internal consistency (Cronbach’s alpha = 0.82) and construct validity, including convergent and discriminant validity, in child samples [36]. This measure has primarily been used in samples of parents with children ages six to nine years. In the current study, the Gratitude Moments Questionnaire demonstrated good internal consistency (Cronbach’s alpha = 0.86).

### 2.2. Statistical Analysis

Internal consistency reliability was determined by the Cronbach’s alpha statistic. For the purpose of the current study, the strength of Cronbach’s alpha coefficients were determined as follows: >0.9, excellent; >0.8, good; >0.7, acceptable; >0.6, questionable; >0.5, poor; and <0.5, unacceptable [25].

An intraclass correlation coefficient (ICC) was used to determine the test–retest reliability of the EES-C parent version. Specifically, a two-way mixed-effects model was utilized that assumes a fixed rater (parent) effect and random error [26]. ICC values were interpreted as follows: <0.5, poor reliability; 0.5–0.75, moderate reliability; 0.75–0.9, good reliability; and >0.9, excellent reliability. Of the 207 parents in the sample, 63 completed the EES-C at least two weeks following the initial administration of the scale to evaluate test–retest reliability.

To assess convergent validity, the correlation between the EES-C Short-Form parent version and a measure of a theoretically related construct were examined. The Child Eating Behavior Questionnaire (CEBQ) was selected as it measures emotional over-eating, among other maladaptive eating behaviors in children (e.g., emotional undereating and satiety responsiveness), that are theoretically related to emotional eating [37,38]. A Pearson correlation was used to assess the relationship between the EES-C and the CEBQ emotional overeating scale. The strength of the correlation was determined following Cohen’s guidelines (small = 0.10, medium = 0.30, large = 0.50) [32].

The present study examined discriminant validity by evaluating a Pearson correlation between the EES-C Short-Form parent version and a measure of gratitude (Gratitude Moments Questionnaire). Limbers and colleagues previously demonstrated that the child-report EES-C Short-Form demonstrated a small, non-significant correlation with gratitude—a theoretically unrelated construct (0.086, *p* = 0.347) [22].

The ICC was used to measure the agreement between child and parent reports of child emotional eating on the EES-C Short-Form for children aged 8 to 12 years. ICC values were interpreted as follows: <0.5, poor agreement; 0.5–0.75, moderate agreement; 0.75–0.9, good agreement; and >0.9, excellent agreement.

A confirmatory factor analysis (CFA) was used to examine the internal structure of the parent-report EES-C Short-Form. A unidimensional model was examined in which all ten items loaded onto a single latent factor. In postulating a single latent factor, the overall tendency towards emotional eating was evaluated, treating the scale items as indicators of a similar underlying construct. The decision for a unidimensional structure is consistent with previous research that suggests the EES-C Short-Form measures a singular dimension [39]. Additionally, previous studies have provided support for the one-factor structure of the EES-C Short-Form self-report version [21,24]. Considering the ordinal nature of the variables comprised on the EES-C, the diagonally weighted least-squares (DWLS) estimator was used.

Following recommendations by Hu and Bentler, the Comparative Fit Index (CFI) and Standardized Root Mean Residual (SRMR) indices were examined to determine the quality of fit for the unidimensional model [31]. The CFI examines how well the unidimensional model fits, relative to a baseline model of independence. Additionally, the CFI is less susceptible to sample-size effects than other indicators of model fit (e.g., the chi-square statistic). Adequate fit, as indicated by the CFI, is a value greater than or equal to 0.95 [31]. The SRMR examines the standardized differences between the observed and predicted covariance matrices. Adequate fit, as indicated by the SRMR, is a value less than or equal to 0.09 [31].

The chi-square statistic was examined, though interpreted with caution, as this statistic can be overly sensitive in larger samples. This tendency can lead to the rejection of potentially well-fitting models when sample sizes are large [40,41]. Considering this tendency, the CFI and SRMR were examined more heavily to determine model fit, as they are both more robust to sample-size effects and nuanced in perspective of model fit.

To measure the differences between children classified as being overweight or obese and those classified as having a healthy weight (as designated by the Centers for Disease Control and Prevention’s BMI for age growth charts) as an indicator of construct validity, known-groups analyses were conducted [42]. Independent samples *t*-tests were used to investigate whether there were statistically significant differences between parent-reported levels of their child’s emotional eating between children who were overweight/obese (*n* = 46) and healthy-weight children (*n* = 99). To determine the magnitude of differences between groups, Cohen’s d effect sizes were calculated. Effect sizes for differences in means were classified according to Cohen’s recommendations, where 0.2 is small, 0.5 is medium, and 0.8 is considered a large magnitude of difference [32].

Data analyses were performed using IBM SPSS© Statistics© for Mac, Version 29, and R©, Version 1.4.1106, using the lavaan and semPlots packages.

## 3. Results

### 3.1. Participant Characteristics

Data from 16 parent and 16 child participants were excluded from the study due to a large number of missing responses. Descriptive statistics for the final sample of 207 parents and 144 children and study measures are presented in Table 1, Table 2 and Table 3. Parent participants ranged in age from 26 to 69 years (*M* = 39.72, *SD* = 7.76). As well as the 207 parents who participated in the study, 144 children also participated. Most of the parent sample identified as female (86.5%), with a smaller percentage identifying as male (12.6%) or “other” (0.5%). The composition of parental race and ethnicity consisted of 72.5% White, 13% Latino, 6.8% Black, 5.8% Asian, and 2.4% “other”, which included those who identified as biracial or multiracial. The sample was highly educated, with 56.1% indicating a bachelor’s degree or higher. In terms of employment status, 71.5% of the sample endorsed that they work full-time, 22.7% worked part-time, and 3.86% of the sample endorsed that they were not currently employed.

Child participants ranged in age from 5 to 12 years (M = 9.37, SD = 2.05). The majority of the child sample was female (52.7%), with a slightly smaller percentage identifying as male (47.3%). The composition of child race and ethnicity consisted of 65.9% White, 13.1% Latino, 6.9% Black, 4.1% Asian, and 7.6% “other”, which includes those who identified as biracial or multiracial. Most of the child sample was classified as having a healthy weight (47.83%), while smaller percentages were classified as being underweight (3.38%), overweight (13.53%), and obese (8.69%). Per the parent report, the majority of the child sample did not have a pre-existing mental health diagnosis (70%). The number of mental health diagnoses endorsed by parents ranged from one to four diagnoses, with 20% of the sample endorsing one diagnosis, 5.8% endorsing two diagnoses, 2.9% endorsing three diagnoses, and 0.5% endorsing four diagnoses. Attention-deficit/hyperactivity disorder was the most reported diagnosis (29.5%). Additional descriptive statistics for the child sample are presented in Table 2.

### 3.2. Internal Consistency Reliability

The parent version of the EES-C Short-Form demonstrated excellent internal consistency, as measured by the Cronbach’s alpha statistic, with a coefficient of 0.94.

### 3.3. Test–Retest Reliability

There was a significant correlation between the parent’s initial rating of their child’s emotional eating on the EES-C Short-Form and their rating two weeks later (ICC = 0.56, *p* < 0.001).

### 3.4. Convergent Validity

The Pearson correlations between the parent version of the EES-C Short-Form and CEBQ scales are contained in Table 4. The parent-report EES-C Short-Form demonstrated a small, statistically significant correlation (*r* = 0.25, *p* < 0.001) with the CEBQ emotional overeating scale. The correlations between the parent version of the EES-C Short-Form and other scales comprising the CEBQ ranged from 0.04 to 0.41. There was a medium, significant correlation between the EES-C Short-Form and the enjoyment of food scale (*r* = 0.41, *p* < 0.001).

### 3.5. Discriminant Validity

The parent version of the EES-C Short-Form demonstrated a small, positive, non-significant correlation with a measure of parent-reported child gratitude (*r* = 0.07, *p* = 0.30).

### 3.6. Agreement

There was a small correlation between parent- and child-rated emotional eating on the EES-C Short-Form (ICC = 0.31, *p* < 0.001). The magnitude of the correlation was considered weak, indicating poor agreement between parents and children [26].

### 3.7. Model Fit

A unidimensional model was evaluated in which all items on the scale were loaded onto a single latent factor—the overall tendency towards emotional eating. Considering the ordinal nature of the variables comprised on the EES-C Short-Form, the diagonally weighted least-squares (DWLS) estimator was used. The chi-square statistic produced a significant value (χ^2^ (35) = 124.00, *p* < 0.001). The CFI value of 0.997 and SRMR value of 0.046 revealed that the one-factor model met the criteria for an excellent fit (CFI = 0.997, SRMR = 0.046), using the recommendations of Hu and Bentler [31]. Standardized factor loadings are contained in Table 5. Standardized loadings ranged from 0.58 to 0.90 onto the unidimensional factor. Notably, one item (Item 1) demonstrated a standardized factor loading below 0.70 (0.58)—a commonly suggested threshold in determining meaningful contribution to a scale [43].

### 3.8. Known Groups

There was a small, non- significant difference in parent-reported emotional eating on the EES-C Short-Form in healthy-weight children (*M* = 22.12) and children who were overweight or obese (*M* = 22.58; *d* = −0.06, *t*(145) = −0.32, *p* = 0.98).

## 4. Discussion

The purpose of this study was to evaluate the psychometric properties of a parent-report version of the EES-C Short-Form in children between the ages of 5 and 12 years. The hypothesis that the EES-C would demonstrate adequate internal consistency reliability was supported. This finding is consistent with the preliminary study evaluating the measure’s internal consistency reliability [24]. There was a significant correlation between parents’ initial ratings of their child’s emotional eating on the EES-C and their ratings two weeks later, supporting the measure’s test–retest reliability [26]. This finding is in line with a study conducted by Tanofsky-Kraff and colleagues on the EES-C self-report original version, which found intraclass correlation coefficients ranging from 0.59 to 0.74 in an adolescent sample, demonstrating moderate test–retest reliability [16]. The finding that the parent version of the EES-C Short-Form demonstrated moderate test–retest reliability suggests that the measure can be utilized consistently over time, allowing for the longitudinal tracking of emotional eating. This is noteworthy because it may enable mental health and medical professionals to monitor changes in emotional eating patterns and assess the effectiveness of interventions.

The EES-C was significantly correlated to theoretically related dimensions of maladaptive eating behaviors, consistent with our hypothesis and supporting the measure’s convergent validity. These findings are consistent with the preliminary study evaluating the parent-report version of the EES-C Short-Form, which found that maternal reports of their child’s emotional eating were associated with maternal concern about their child’s weight (*r* = 0.285; *p* < 0.001), restricted eating (*r* = 0.238; *p* < 0.001), and monitoring their child’s food intake (*r* = 0.136; *p* < 0.05) [24]. Interestingly, in the current study, a medium significant correlation was demonstrated between the EES-C and the enjoyment of food scale on the CEBQ. Many individuals endorse an improvement in mood following eating, as consuming good-tasting food may naturally be reinforcing [44]. Considering this, individuals who particularly enjoy food may be more likely to eat to resolve unpleasant emotions. Additionally, the EES-C original version has historically demonstrated good convergent validity with scales measuring maladaptive eating behaviors, such as Loss of Control (LOC) episodes [16,39]. The hypothesis that the EES-C Short-Form would demonstrate a small, non-significant correlation with a scale measuring gratitude was supported. This finding is in line with a previous finding (Limbers et al., 2020) [22] and demonstrates that the EES-C is not associated with a theoretically dissimilar construct, supporting the measure’s discriminant validity. The original EES-C has demonstrated mixed findings with regard to discriminant validity [16,45]. Specifically, Perpiña et al. demonstrated moderate correlations between the EES-C Spanish version and the Children’s Depression Inventory (CDI) and Child Behavior Checklist (CBCL), indicating acceptable discriminant validity [45]. Conversely, Tanofsky-Kraff and colleagues reported the low discriminant validity of the original EES-C with measures of trait anxiety and externalizing behaviors [16]. Possible explanations for the differences in findings are that the studies were conducted using different-language versions of the EES-C and in clinical versus community samples.

Not consistent with our hypothesis, agreement between children and parents when rating child emotional eating on the EES-C Short-Form was considered weak [26]. While no previous studies have evaluated agreement between parent- and child-rated emotional eating, our finding is consistent with the literature demonstrating poor agreement between children and parents in terms of children internalizing symptoms and more general disordered-eating behaviors [46]. One plausible explanation for the current finding is that children who engage in high levels of emotional eating may be prone to underreporting these behaviors due to feelings of shame and guilt associated with having an elevated weight status or engaging in maladaptive eating behaviors [47]. This may have contributed to greater discrepancies between parent and child reports of child emotional eating in the present study.

Supporting our hypothesis, the unidimensional model demonstrated excellent fit. This is consistent with previous findings evaluating the EES-C Short-Form self- and parent-report versions [21,24]. This finding supports that the unidimensional structure accurately represents the relationships among the observed variables, supporting the scale’s structural validity. Additionally, the model fit ensures that the scale aligns with theoretical models of emotional eating in children. This finding supports the claim that the EES-C Short-Form captures a singular underlying construct—the tendency to emotionally eat—which reinforces the measure’s theoretical foundation [39]. Despite the model demonstrating excellent fit, standardized factor loadings revealed one item (Item 1) whose standardized loading was below a suggested magnitude for determining meaningful contribution to a scale (0.70) [43]. Item 1 asks parents to determine whether their child emotionally eats when they feel they are not doing enough. While the preliminary study examining model fit found acceptable standardized factor loadings on all items, Item 1 also demonstrated the lowest factor loading (0.73), just above the recommended threshold for meaningful contribution [24]. Our findings suggest that Item 1 may demonstrate less-than-ideal efficacy in contributing to the scale’s effective measurement of emotional eating. Future studies examining the psychometric properties of the parent version of the EES-C Short-Form should clarify the degree to which Item 1 contributes effectively to the overall measurement of emotional eating from the perspective of parents. With additional research, consideration may be given to revising or eliminating Item 1 to further enhance the structural validity of the parent-report EES-C short-form.

Contrary to our hypothesis, parents of children with elevated weight status were not more likely to endorse emotional eating compared to parents of children classified as having a healthy weight. A preliminary study found that parents were more likely to endorse emotional eating in children with elevated weight status or who were obese than parents of healthy-weight children [24]. In this previous study, the weight of the child was subjectively reported by the mother. In the current study, child weight was collected objectively by a trained research assistant. A child’s weight fluctuates and changes during childhood [48]. It is possible that parents are not accurate subjective reporters of their children’s weight [49], and their reports of child weight in the previous study may also be impacted by their perception of their child and whether they perceive their child to be an emotional eater [50]. One possible explanation for inconsistencies in findings about emotional eating and weight status is that during childhood, emotional eating may act as a predictive factor for the development of elevated weight status [51], though children who emotionally eat may not consistently have elevated weight status [13]. While our study did not explicitly assess whether there were differences in emotional eating between boys and girls on the EES-C Short-Form, given the well-documented differences in emotional eating and emotion dysregulation between males and females [52], future research using the EES-C Short-Form parent version should evaluate whether there are differences in how parents rate their child’s emotional eating based on child gender.

The early identification of children who emotionally eat is important for several reasons. First, the ability to identify children who are rated highly on scales of emotional eating may increase early intervention and prevention strategies, such as parent-guided mindfulness [53]. These early interventions may lead to prevention efforts to disrupt patterns of emotional eating. Preventative efforts may also reduce the risk of developing well-established adverse health outcomes commonly associated with emotional eating during adolescence and adulthood [9,54]. Second, assessing emotional eating in children allows for a more comprehensive understanding of the child’s psychological well-being and coping skill development and utilization. Emotional eating is associated with underlying mood and anxiety concerns, as well as psychological distress [55]. Thus, early intervention could improve children’s quality of life and psychological well-being. Considering these impacts, it is important to have well-established measures of emotional eating in children.

While self-report is generally considered the standard in the measurement of emotional eating for children ages 8 years and older, young children are less able to differentiate internalizing states and identify their motives for eating, particularly when introspection is required to identify the antecedent of their eating behaviors [23]. Despite a clear need, there was not a well-validated parent-report measure of emotional eating available for use until the parent-report EES-C Short-Form was developed (Summers and Limbers, 2024). Taken as a whole, the current study supports the psychometric properties of the parent version of the EES-C Short-Form in children 5 to 12 years.

There were several limitations that should be considered in interpreting the findings. The sample population was largely homogenous in terms of race and ethnicity, with a large majority of participants identifying as White and non-Hispanic. Considering this, the results of the current study may not generalize to the general population of children. Future studies would benefit from utilizing a more diverse sample considering emotional eating may be more prevalent in certain cultural contexts [24]. Similarly, the sample was homogenous in terms of child weight status, with the majority of the sample classified as having a healthy weight. The prevalence of obesity for children ages 6 to 11 years old is approximately 20.7% in the United States (CDC, 2022); however, the prevalence of children classified as having obesity in the current sample was only 8.69%. Future studies would benefit from a child sample that is more representative of the general population in terms of weight status to increase generalizability and further elucidate the relationship between weight status and emotional eating. We did not apply a correction for multiple comparisons or control for covariates in our analysis. There was not a large enough sample within each age group to analyze differences in results based on the age of the child participants. It is possible that results from the current study may differ based on the age of the child. In a post hoc analysis, we found small, non-significant associations between child age and parent- (*r* = 0.08) and child-rated (*r* = 0.04) emotional eating on the EES-C Short-Form, which suggests there may have been few differences if the data were analyzed separately by child age. Children completed the study questionnaires in a classroom alongside other participating children. It is possible that children provided socially desirable responses resulting from uncertainty or fear that their peers could view and judge their responses to questions. This could have resulted in the underreporting of behaviors or feelings, thus impacting the validity of responses. Considering this limitation, future studies may benefit from asking children to complete questionnaires in a confidential space to reduce the risk of response bias.

## 5. Conclusions

In conclusion, the current study found evidence to support the parent-report version of the EES-C Short-Form’s internal consistency reliability, test–retest reliability, convergent and discriminant validity, and structural validity. The absence of a well-established parent-report measure of emotional eating in children has been a gap in the literature, and the introduction and validation of this measure has implications in clinical practice and research. For the purpose of screening children in school or primary care settings, where there is limited time and availability of resources, this measure would be practical and helpful in identifying children who may be at risk of developing adverse health outcomes or more-severe eating disorder pathology. The limited time and resources often encountered in these settings require efficient and effective screening measures. The brevity of the parent-report EES-C Short-Form would be well-suited for school and primary care settings as well as clinical research settings as it can be administered quickly, making it a feasible option for when time constraints exist.

## Figures and Tables

**Table 1 nutrients-16-03030-t001:** Parent sample characteristics.

Characteristic	*N*	Percent of Sample
Gender of Parent		
Male	26	12.6
Female	179	86.5
Other	1	0.5
Missing	1	0.5
Age		
20 to 30 years	11	5.3
31 to 40 years	80	38.6
41 to 50 years	73	35.2
51 to 60 years	4	1.9
61 to 70 years	3	1.4
Missing	36	17.3
Body Mass Index		
Underweight	1	0.5
Healthy weight	74	35.7
Overweight	53	25.6
Obese	62	30.0
Missing	17	8.2
Parental Level of Education		
High School	13	6.3
Some college	33	16.1
Four-year college	69	33.7
Master’s degree	50	24.4
Doctoral or Professional degree	28	13.7
Doctoral academic degree	11	5.3
Missing	3	1.0
Employment Status		
Full-time	148	71.5
Part-time	47	22.71
None	8	3.86
Characteristic	N	Percent of Sample
Missing	4	1.93
Race/Ethnicity		
Asian	11	5.3
Black	14	6.8
Hispanic	27	13.0
White	150	72.5
Other	5	2.4

*Note. N* = 207. Participant age ranged from 26 to 69 years old with a mean age of 39.72 years (*SD* = 7.76 years).

**Table 2 nutrients-16-03030-t002:** Child sample characteristics.

Characteristic	*N*	Percent of Sample
Gender of Child		
Male	98	47.3
Female	109	52.7
Age		
5 years	4	1.9
6 years	15	7.2
7 years	28	13.5
8 years	31	15.0
9 years	23	11.1
10 years	32	15.5
11 years	30	14.5
12 years	44	21.3
Body Mass Index		
Underweight	7	3.38
Healthy weight	99	47.83
Overweight	28	13.53
Obese	18	8.69
Missing	55	26.57
Race/Ethnicity		
Asian	6	4.1
Black	10	6.9
Hispanic	19	13.1
White	95	65.9
Other	11	7.6
Missing	3	2.0
Characteristic	N	Percent of Sample
Number of Diagnoses		
0	146	70.5
1	42	20.3
2	12	5.8
3	6	2.9
4	1	0.5

*Note. N* = 144. Participant age ranged from 5 to 12 years old with a mean age of 9.37 years (*SD* = 2.05 years).

**Table 3 nutrients-16-03030-t003:** Descriptive statistics for questionnaires.

Instrument	*N*	Mean	SD	Cronbach’s Alpha
EES-C, Parent	200	22.34	8.29	0.94
EES-C, Self	132	21.93	6.60	0.88
CEBQ	202			
Food Responsiveness		13.38	6.39	0.34
Emotional Overeating		2.15	0.77	0.62
Enjoyment of Food		15.15	3.05	0.89
Desire to Drink		8.56	3.06	0.71
Satiety Responsiveness		14.17	3.46	0.79
Slowness in Eating		8.45	2.48	0.70
Emotional Undereating		11.23	3.37	0.76
Food Fussiness		14.37	2.77	0.53
Gratitude Moments	203	22.90	7.06	0.86

**Table 4 nutrients-16-03030-t004:** Pearson correlations of the CEBQ subscales and EES-C-P.

	FR	EOE	EF	DD	SR	SE	EUE	FF	EEC-C-P
1. Food Responsiveness	1								
2. Emotional Overeating	0.40 **	1							
3. Enjoyment of Food	−0.52 **	0.34 **	1						
4. Desire to Drink	0.12	0.22 **	0.14	1					
5. Satiety Responsiveness	−0.34 **	−0.14 *	−0.57 **	0.02	1				
6. Slowness in Eating	−0.11	−0.04	−0.37 **	0.03	0.47 **	1			
7. Emotional Undereating	0.08	0.37 **	−0.02	0.20*	0.20 *	0.17 *	1		
8. Food Fussiness	−0.14 *	−0.09	−0.52 **	0.09	0.36 **	0.26 **	0.11	1	
9. EES-C-P	0.25 **	0.25 **	0.41 **	0.04	−0.23 **	−0.17	0.13	−0.19	1

* *p* < 0.05; ** *p* < 0.01.

**Table 5 nutrients-16-03030-t005:** Standardized factor loadings.

Item	Factor Loading
1. Not doing enough	0.576
2. Disobedient	0.841
3. Sad	0.898
4. Jealous	0.903
5. Lonely	0.801
6. Confused	0.893
7. Nervous	0.799
8. Angry	0.883
9. Guilty	0.903
10. Helpless	0.857

*Note.* Items are listed in the same order as they appear on the parent-report EES-C Short-Form.

## Data Availability

These data will be made available upon reasonable request to the first author.

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
