# Peer review of "Evaluating Emotional Eating in Children from the Perspective of Parents: Psychometric Properties of the Parent Version of the Emotional Eating Scale Adapted for Children and Adolescents"

_nutrients, 2024, doi:10.3390/nu16173030_

Round 1

Reviewer 1 Report

Comments and Suggestions for Authors

The paper was well-presented and coherent with analyses all reported clearly. On that basis, I only have a few minor comments that the authors could address to improve the quality of the paper. 

1. This is a minor point, but lines 49 to 53 of the introduction are a bit convoluted and the authors could just report that the size of the correlation between the constructs to demonstrate convergent validity

2. In the methods, the line 'The CEBQ's subconstructs are theoretically related to emotional eating' is a bit unclear - presumably the authors mean some of the subscales rather than all of them, so could be more specific/precise here

3. This is very picky of me, but if two constructs are theoretically unrelated this is likely to be based on an understanding of what those constructs are rather than a lack of correlation in a previous study (line 210) so would reword this sentence to clarify that they are theoretically unrelated and that in support of this. a previous study has reported a small non significant relationship

4. One line 240 please reference the guidelines used to categorise children into the different weight categories (which presumably invole adjustment for age of the child)

5. In lines 356 to 360 of the discussion, the reasons for discrepancies could be discussed in a bit more detail rather than simply described. I am not surprised that depression scores correlated with emotional eating for example. The authors could reflect a little more on why there are differences in findings, taking into account that the magnitude of the correlation coefficient is a more interesting indicator of discriminant validity, rather than its statistical significance

6. In line 366, could the authors expand a little on the point about shame, based on the findings from the previous study? 

7. Very minor point, but the section on implications starting with line 407 - it might be useful to start with why emotional eating is problematic (which comes a couple of sentences after) and then move onto implications for prevention

8. Line 431, do the authors have statistics on the prevalence of overweight and obesity in those aged 5 to 12 rather than 2 to 19? If so, it would be a closer comparison group

Author Response

Please see attached file as supplementary materials. The responses to the reviewers have been detailed here.

Reviewer 2 Report

Comments and Suggestions for Authors

This original article provides a preliminary validation of the parent version of the Emotional Eating Scale Adapted for Children and Adolescents (EES-C) Short-Form. While the results are noteworthy and may have clinical implications, the study requires major revisions to enhance the findings due to methodological concerns, partial limitations, and a lack of in-depth discussion of existing literature, as detailed below.

Thus, all points listed below need to be addressed.

ABSTRACT

Please mention the limitations in the abstract.

INTRODUCTION

The introduction is generally appropriate. However, to better highlight the rationale of the study, it is crucial to strengthen the link between emotional eating and emotion dysregulation. The clinical relevance of emotional eating (EE) may be mediated by its association with emotion dysregulation, extending beyond childhood into adulthood. Recent literature has found that emotion dysregulation links to pathological eating styles and psychopathological traits in bariatric surgery candidates, making it a crucial transdiagnostic symptom for a variety of psychiatric disorders, as well as a risk factor for somatic conditions and pro-inflammatory status. Up-to-date literature should be included to support this point.

METHODS

Please provide further details on corrections for multiple comparisons, adjustment for covariates (e.g., sex, socioeconomic status, comorbidities, medication, and/or education), or include these as limitations.

Although age could not be accounted for, this is correctly listed as a limitation. To partly address this, please provide additional details on correlations between age and the clinical scores acquired (this can be included in a supplementary section).

RESULTS

Please provide further details on the screening of participants, including the number and reasons for the exclusion of participants.

DISCUSSION

The discussion provides interesting information overall. However, gender is a significant determinant of differences in emotion regulation and emotional eating. While the sample was generally matched for gender, this aspect should be expanded in the discussion, with reference to up-to-date literature on the role of gender (e.g., “Emotional Regulation Underlies Gender Differences in Pathological Eating Behavior Styles of Bariatric Surgery Candidates”).

CONCLUSIONS

Please expand the conclusions to improve the discussion of clinical and research perspectives and implications.

LIMITATIONS

Please expand the limitations section to include the relevant points mentioned above.

SUPPLEMENTARY MATERIALS

NA

Comments on the Quality of English Language

Please correct typos throughout the manuscript.

Add an explanation of abbreviations throughout the manuscript upon first use, and ensure that figures and tables are interpretable independently from the text.

Author Response

(The authors gave the same response as above.)

Round 2

Reviewer 2 Report

Comments and Suggestions for Authors

All comments have been addressed or acknowledged as limitations. I am pleased to recommend this article for publication.

Comments on the Quality of English Language

Overall ok

Author Response

Thank you for this feedback.  The limitations at the end of the abstract were added to address one of the reviewer's comments.  We have now removed these limitations from the abstract and added some language related to clinical implications. We also now use first person language with respect to obesity throughout the manuscript. Changed in the manuscript are denoted in red font.
